# Immunoexpression of stem cell markers SOX-2, NANOG AND OCT4 in ameloblastoma

Karolyny Martins Balbinot[1], Felippe José Almeida Loureiro[2],
Giordanna Pereira Chemelo[2], Ricardo Alves Mesquita[3],
Aline Maria Pereira Cruz Ramos[4], Rommel Thiago Jucá Ramos[5],
Artur Luiz da Costa da Silva[5], Sílvio Augusto Fernandes de Menezes[6],
Maria Sueli da Silva Kataoka[2], Sergio de Melo Alves Junior[1] and
João de Jesus Viana Pinheiro[2]

[1] Laboratory of Pathological Anatomy and Immunohistochemistry, Federal University of Pará,
Belém, Pará, Brazil
[2] Cell Cultivation Laboratory, Federal University of Pará, Belém, Pará, Brazil
[3] Department of Oral Surgery and Pathology, Federal University of Minas Gerais, Belo Horizonte,
Minas Gerais, Brazil
[4] Health Science Institute, Federal University of Pará, Belém, Pará, Brazil
[5] Biological Engineer Laboratory, Park of Science and Technology, Belém, Pará, Brazil
[6] Department of Periodontics, Pará State University, Belém, Pará, Brazil

Corresponding author
João de Jesus Viana Pinheiro,
radface@hotmail.com

## ABSTRACT

**Background:** Ameloblastoma (AME) is characterized by a locally invasive growth pattern. In an attempt to justify the aggressiveness of neoplasms, the investigation of the role of stem cells has gained prominence. The SOX-2, NANOG and OCT4 proteins are important stem cell biomarkers.

**Methodology:** To verify the expression of these proteins in tissue samples of AME, dentigerous cyst (DC) and dental follicle (DF), immunohistochemistry was performed and indirect immunofluorescence were performed on the human AME (AME-hTERT) cell line.

**Results:** Revealed expression of SOX-2, NANOG and OCT4 in the tissue samples and AME-hTERT lineage. Greater immunostaining of the studied proteins was observed in AME compared to DC and DF ($p < 0.001$).

**Conclusions:** The presence of biomarkers indicates a probable role of stem cells in the genesis and progression of AME.

## INTRODUCTION

Ameloblastoma (AME) is an odontogenic tumour of epithelial origin (*Effiom et al., 2018*) and, although classified as a benign tumour, it is characterized by a locally invasive growth pattern, which can reach large proportions and promote facial deformities in patients (*Rioux-Forker et al., 2019*).

Surgical removal is the treatment of choice, but when conservative techniques are applied, small islands of tumour are not completely removed, leading to local recurrence in 60–80% of cases of solid AME (*Gomes et al., 2010*).

In the search for cellular mechanisms that justify the local aggressiveness of this benign lesion, the investigation of the role of stem cells (SCs) and cancer stem cells (CSCs) has gained prominence in tumour biology, with research identifying their participation in growth, angiogenesis, progression, tumour recurrence and self-renewal potential (*Reya et al., 2001*; *Silva et al., 2016*; *Pardal, Clarke & Morrison, 2003*).

The SOX-2, NANOG and octamer-binding protein 4 (OCT4) are important biomarkers in the analysis of the presence of SCs. They act as critical regulators of embryonic self-renewal and pluripotency capable of mediating tumour proliferation and differentiation (*Chambers et al., 2003*; *Luo et al., 2013*; *Ren, Zhang & Ji, 2016*).

Furthermore, the relation of these proteins seems to play an oncogenic role, considering studies that point to the presence of these factors in different types of cancers, such as lung adenocarcinoma, breast, colorectal and gastric cancer (*Luo et al., 2013*; *Ren, Zhang & Ji, 2016*).

The SOX-2 protein (HMG-box gene 2 related to SRY) acts as an important transcription factor in maintaining the self-renewal capacity of SCs. Previous studies have shown it to be associated with a pro-oncogenic function (*Wilbertz et al., 2011*), in AME (*Juuri et al., 2013*) and ameloblastic carcinoma (*Khan et al., 2018*; *Lei et al., 2014*), with a differentiated expression between the last two.

NANOG (homeodomain protein) is another transcription factor that plays a central role in maintaining cell pluripotency during embryonic development, in addition to being associated with cell proliferation and renewal (*Fan et al., 2017*). The high expression of NANOG has also been identified in patients with some types of cancer (*Luo et al., 2013*; *Yu & Cirillo, 2020*), but it has not yet been studied in AME.

OCT4 also acts in pluripotency and cell self-renewal (*Schulenburg et al., 2006*), and has been associated with cell proliferation and tumour progression (*Hu et al., 2020*). The expression of OCT4 in AME (*Monroy et al., 2018*) was related to cell development and differentiation, and an initial study on the expression of this protein in lesions of odontogenic origin showed divergence in expression between AME and ameloblastic carcinoma (*Khan et al., 2018*).

SOX-2 and OCT4 are considered essential regulators for the maintenance and early development of SCs (*Chambers et al., 2003*). Although they have independent roles in different cell types, they present a synergistic interaction that leads to the transcription of target genes (*Avilion et al., 2003*), with NANOG as one of the targets of this interaction (*Rodda et al., 2005*).

When together, SOX-2, NANOG and OCT4 bind to the promoters of their own genes, forming interconnected self-regulatory loops (*Boyer et al., 2005*). It is believed that this self-regulation network can provide advantages for SCs, such as reduced response time to environmental stimuli and greater stability of gene expression, thus maintaining cell fate (*Rodda et al., 2005*). These characteristics are important for cell survival, stability and tumour progression.

It has been reported that odontogenic neoplasms, such as AME, originate from remaining SCs from the dental lamina (Harada et al., 2002). However, the true contribution of SCs to the molecular mechanism involved in the pathogenesis of AME still needs clarification (Silva et al., 2016; Juuri et al., 2013). This is the first work to study these three proteins (SOX-2, NANOG and OCT4) in AME. In this sense, identifying the proteins that are related to cellular self-renewal and pluripotency may justify the biological behaviour of AME and becomes extremely important to the development of better treatments and prognosis.

## MATERIALS AND METHODS

### Sample

This was an experimental laboratory study. For the *in vivo* study, 23 AME samples were used, retrieved from the archives of the Department of Oral Pathology, Faculty of Dentistry, University Centre of the State of Pará (CESUPA). DC, as well as AME, is derived from the odontogenic epithelium, but has a less aggressive behaviour. Thus, 10 samples of DC were used as a control, added to 10 samples of DF (tissue without neoplastic alterations of odontogenic origin), obtained from the Laboratory of Pathological Anatomy and Immunohistochemistry, Faculty of Dentistry, Federal University of Pará (UFPA).
The clinical data of the AME samples were acquired through medical records with reports present in the files, collected manually, and histologically classified by two oral pathologists. For the *in vitro* study, the cell line derived from human AME, called AME-hTERT, established at the Cell Culture Laboratory of the Faculty of Dentistry, Federal University of Pará (UFPA) (Cruz et al., 2021), was used. This study was registered and approved by the Human Research Ethics Committee of the Health Sciences Institute of the Federal University of Pará—CEP/ICS/UFPA (CAAE: 30647720.6.0000.0018). Informed consent was waived by this one.

### Immunohistochemistry

Immunohistochemical analysis was performed according to Mitre et al. (2021), where AME, DC and DF tissues were incubated with Anti-SOX-2 (1:50 Sigma, St. Louis, MO, USA), Anti-Nanog (1:150 Millipore, Burlington, MA, USA) and Anti-Oct4 (1:25 Millipore, Burlington, MA, USA) antibodies for 1 h. As a positive control, samples of oral squamous cell carcinoma were used and as a negative control, the primary antibody was replaced by BSA and fetal bovine serum in TRIS buffer.

### Immunohistochemical evaluation

Five brightfield images were randomly acquired from each AME, DC and DF of regions with intact epithelium, and in the case of AME that was representative of the lesion, a sample on an AxioScope microscope (Carl Zeiss, Oberkochen, Germany, DEU) equipped with an AxioCam HRC colour CCD camera (Carl Zeiss, Oberkochen, Germany) was obtained. Images were taken at 400× magnifications and saved in TIFF format. The areas stained with diaminobenzidine were analysed using the "Immunohistochemistry (IHC) Image Analysis Toolbox" of the ImageJ software (National Institute of Mental Health

(NIMH), National Institute of Health (NIH, Bethesda, MD, USA). Semi-automatic image analysis was then performed by detecting DAB staining. The mean percentages of marking obtained in the tumour parenchyma, of five fields per sample, were analysed using the GraphPad Prism 8 software (GraphPad Software Inc., San Diego, CA, USA).

## Cell cultivation

The ameloblastoma cell line was cultured and maintained in culture bottles according to *Valladares et al. (2021)*.

## Indirect immunofluorescence

The AME-hTERT strain was seeded on glass coverslips in 24-well plates and submitted to an indirect immunofluorescence protocol to detect the expression of SOX-2, NANOG and OCT4. This process was initiated by fixing the cells in 2% paraformaldehyde for 10 min, followed by washing with PBS, permeabilization of the membrane with 0.5% Triton X-100 (Sigma, St. Louis, MI, USA) solution for 5 min, a second washing with PBS, and incubation in PBS/BSA (BSA, Sigma, St. Louis, MI, USA) at 1% for 30 min. Subsequently, primary antibodies diluted in PBS/BSA at 1% were incubated for a maximum of 18 h in a humid chamber at 4 °C. The primary antibodies used were: Anti-SOX-2 (1:50 Sigma, St. Louis, MI, USA), Anti-Nanog (1:50 Millipore, Burlington, MA, USA), and Anti-Oct4 (1:50 Millipore, Burlington, MA, USA). To detect the primary antibody, incubation in a solution containing the secondary antibody conjugated to AlexaFluor 488 (Invitrogen, Carlsbad, CA, USA) for 1 h in a dark, humid chamber at room temperature was performed.
For better visualisation of the cytoskeleton, Alexa Fluor 568 Phalloidin (Life Technologies, Carlsbad, CA, USA) was used. The nuclei were labelled with DAPI coupled to ProLong Gold antifade reagent (Invitrogen, Carlsbad, CA, USA). After mounting, the coverslips were analysed in a fluorescence microscope (AxioScope.A1; Zeiss, Jena, Germany), equipped with a digital camera (AxioCamMRc; Zeiss, Jena, Germany), with which images were obtained from the slides to record immunoexpression. All images were acquired at the same magnification (40× lens). The slides were examined under a fluorescence microscope (Axio Scope.A1; Zeiss, Jena, Germany, TH, DEU) equipped with a digital camera (AxioCam MRc; Zeiss, Jena, Germany), in which five images were randomly obtained from each slide for the acquisition of 50 cells per group. All images were acquired at the same magnification (40× objective). The immunostaining evaluation was performed using the ImageJ software.

## Statistical analysis

Clinicohistological data were then tabulated and analysed using descriptive statistics.
To analyse the expression of the three proteins (SOX-2, NANOG and OCT4), comparing the AME samples with those of DC and DF, the statistical tests analysis of variance (ANOVA; samples with parametric distribution) and Kruskal-Wallis were used, followed by Dunn's test multiple comparisons (samples with non-parametric distribution). To verify correlation, the Pearson correlation test (samples with parametric distribution) was used. A significance level of $\alpha = 0.05$ was adopted.

**Table 1 Clinicohistological data of patients with AME.**

| Case | Genre | Age | Localisation | Histological type |
|------|-------|-----|--------------|-------------------|
| 1 | Male | 45 | Upper jaw | Plexiform |
| 2 | Female | 31 | Mandible | Follicular |
| 3 | Male | 31 | Mandible | Plexiform |
| 4 | Female | 27 | Mandible | Plexiform |
| 5 | Male | 34 | Mandible | Plexiform |
| 6 | Female | 32 | Mandible | Follicular |
| 7 | Male | 34 | Mandible | Follicular |
| 8 | Female | 55 | Mandible | Plexiform |
| 9 | Female | 20 | Mandible | Plexiform |
| 10 | Male | 32 | Mandible | Plexiform |
| 11 | Male | 33 | Mandible | Acanthomatous |
| 12 | Female | 59 | Mandible | Follicular |
| 13 | Female | 47 | Mandible | Acanthomatous |
| 14 | Female | 14 | Mandible | Follicular |
| 15 | Male | 42 | Mandible | Granular Cells |
| 16 | Male | 43 | Mandible | Granular Cells |
| 17 | Male | 28 | Mandible | Follicular |
| 18 | Male | 27 | Mandible | Acanthomatous |
| 19 | Female | 62 | Mandible | Follicular |
| 20 | Male | 84 | Upper jaw | Follicular |
| 21 | Male | 49 | Mandible | Follicular |
| 22 | Male | 22 | Mandible | Plexiform |
| 23 | Female | 38 | Mandible | Follicular |

# RESULTS

## Clinicohistological data of patients with AME

In the sample studied, the average age was 39 years, with 61% of individuals below this average and the remaining 39% above. The male gender was observed in 56% of the cases and the female in 44%. The location of greatest involvement was the mandible, totalling 91% of the cases. Regarding the histological type, 10 cases were of the follicular type, eight of plexiform, three acanthomatous and two of granular cells (Table 1).

## AME presents a begreater expression of stem cell markers when compared to dentigerous cyst and dental follicle

AME samples showed higher expressions of SOX-2, NANOG and OCT4 proteins when compared to dentigerous cyst (DC) and dental follicle (DF) ($p < 0.001$). There was no statistical difference between DC and DF ($p > 0.05$) (Fig. 1).

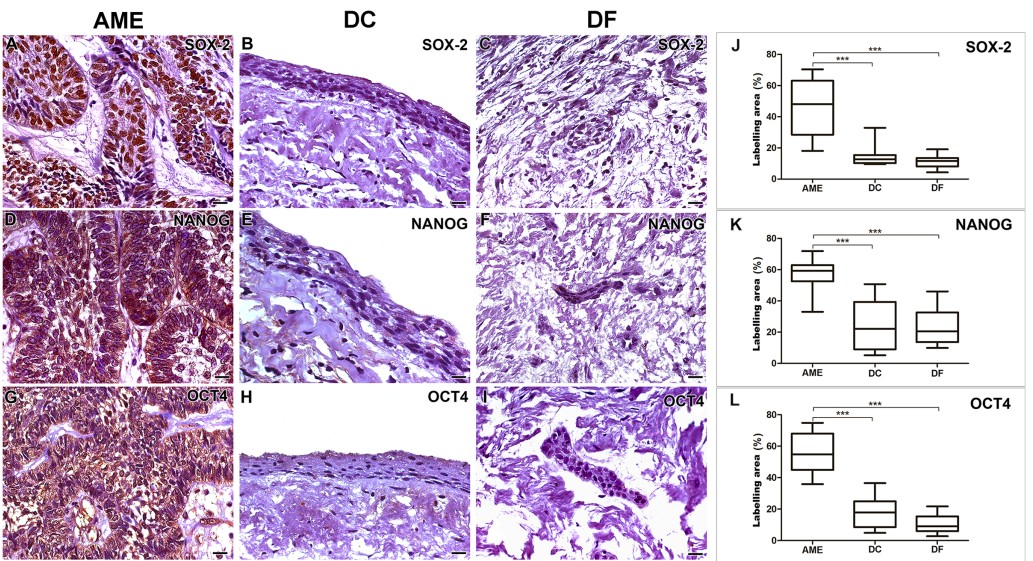

**Figure 1 Comparative immunoexpression of SOX-2, NANOG and OCT4 in AME, DC and DF samples.** Intense imunoexpression of SOX-2 (A), NANOG (D) and OCT 4 (G) were observed in AME samples. There is a predominant nuclear location of immunostaining for the three markers in the parenchymal cells of ameloblastoma. There is a low intensity immunoexpression of the three proteins in the nucleus of some epithelial cells in both DC (B, E and H) and DF (C, F and I) samples. Statistical analysis of percentage of labelling area of the three markers (J, K and L) between AME and DC, and AME and DF. Statistically, a significant difference was observed when comparing the expression of all proteins in AME cells in relation to the expression in DC and DF. (***$p < 0.001$). Scale bar: 20 μm. AME, ameloblastoma; DC, dentigerous cyst; DF, dental follicle.

## Variations in the immunomarking of SOX-2, NANOG and OCT4 in the neoplastic cell compartment of AME

Immunohistochemical staining for SOX-2, NANOG and OCT4 was mainly located in the cords and islands of the odontogenic tumour epithelium. SOX-2 labelling was present predominantly in the nucleus, while NANOG and OCT4 were found in the cell nucleus and diffusely in the cytoplasm of tumour parenchymal cells. Subtle nuclear markings of SOX-2, OCT4 and NANOG were observed in the DC cystic epithelium, and the same occurred in the DF epithelial islands. NANOG was diffusely labelled in the connective tissue of the DF (Fig. 1).

### AME-hTERT lineage presents immunoexpression of stem cell markers

The AME-hTERT strain was verified to express SOX-2, NANOG and OCT4 proteins. Predominantly nuclear expression of SOX-2, NANOG and OCT4 was observed in neoplastic cells (Fig. 2).

## DISCUSSION

This study verified the expression of SOX-2, NANOG and OCT4, SC biomarkers, in AME parenchyma, and subtle expression in the odontogenic epithelium of DC and DF, immunoexpression of the studied proteins was observed in the AME-hTERT cell.

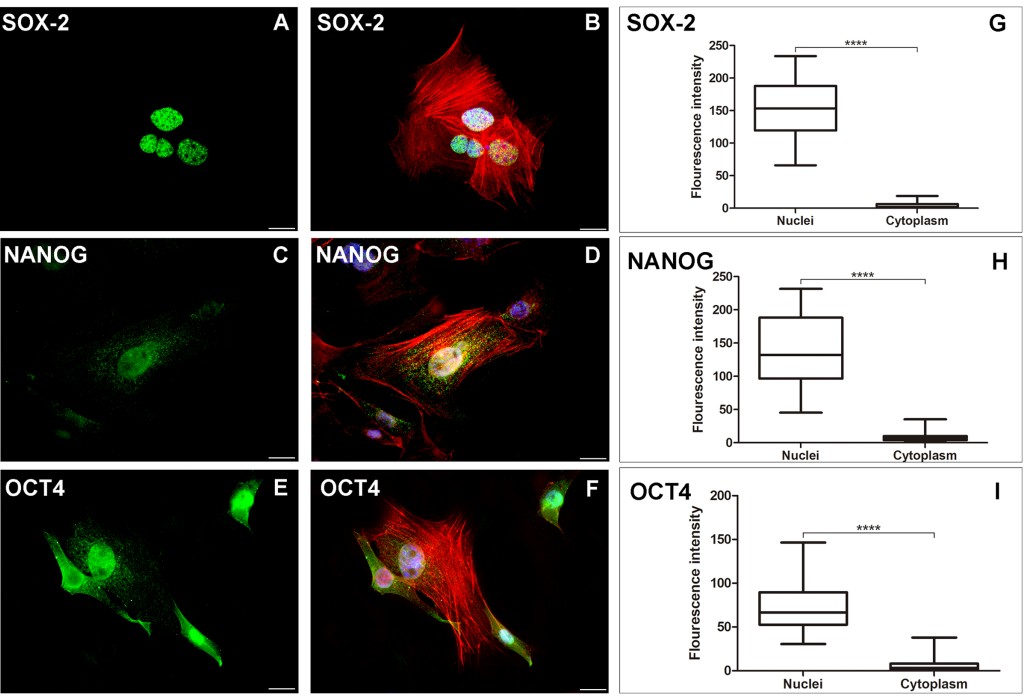

**Figure 2 AME-hTERT cells immunoexpress SOX-2, NANOG and OCT4.** SOX-2 was observed as an intense granular immunostaining (green) distributed mainly in the nuclei (A). Granular expression (green) of NANOG (C) was also observed throughout the nuclei and a weak stain in cytoplasm. OCT4 (E) was observed as granular and dot-like (green) predominantly in the nuclei, but staining in cytoplasm, was also visualized. Cytoskeleton is stained with phalloidin (red) and nuclei is stained with DAPI (blue). Merged staining for SOX-2 (B), NANOG (D), OCT4 (F). Statistical analysis of fluorescence intensity of the three markers studied (G, H and I) between nuclei and cytoplasm of AME-hTERT cells, was carried out. A strong labelling was observed in nuclei when compeer with cytoplasm. Statistically, a significant difference was observed when comparing the expression of all proteins in AME-hTERT cells (****$p < 0.0001$). Scale bar: 20 μm.

The clinical and histological data of the studied samples corroborate the data found in the literature, in which the tumour shows a predominance during the third to seventh decade of life, and the mandible is the location of the highest involvement (*Effiom et al., 2018*).

In our study, we observed a high expression of transcription factors SOX-2, NANOG and OCT4 in the parenchyma of AME samples. On the other hand, there was a weak expression of these factors in DC and DF.

*da Cunha et al. (2013)* performed immunohistochemical analyses of the expression of SCs transcription factors (OCT4, SOX-2, NANOG and Stat-3) in tooth germs, suggesting a wide potential for development and differentiation. In our immunohistochemical analysis, the expression of SOX-2, NANOG and OCT4 was observed both in the periphery and central cells of the nests and epithelial cords, which may be related to the broad expression of these factors in the development of AME, considering that the SCs of the dental lamina are possible targets of carcinogenic agents (*Harada et al., 2002*).

The expression of SOX-2 in this study was predominantly nuclear in the tissue samples of AME and AME-hTERT lineage, as observed in nasopharyngeal carcinoma, gastric,

colorectal, lung and breast cancer (*Luo et al., 2013*; *Ren, Zhang & Ji, 2016*). *Juuri et al. (2013)*, also observed the presence of SOX-2 in AME and related it to the proliferation and embryonic origin of this tumour. In our work, a greater expression of SOX-2 was observed in AME parenchyma in relation to the DC lining epithelium and DF epithelial nests.

Although smaller, the nuclear expression of SOX-2 in the odontogenic epithelium of DC and DF is interesting. *Boumahdi et al. (2014)*, indicate that SOX-2 plays an essential role in tumour genesis, pointing out the role of SOX-2 in regulating functions in the initiation and progression of squamous cell carcinoma. It was indicated that the expression of this protein in cells originating from the odontogenic epithelium of the dental blade could lead to the development of AME (*Juuri et al., 2013*). It was also speculated whether the same could not happen with DC and DF, since their epithelia express SOX-2.

High OCT4 expression was detected in lung adenocarcinoma CSCs (*Hu et al., 2020*) and tumour initiator cells in a mouse model with tumour p53 −/− (*Darini et al., 2012*), depicting that OCT4 expression plays a critical role in tumour cell survival. *Khan et al. (2018)*, did not detect OCT4 expression in 20 cases of aggressive multicystic solid AME, stating that this protein could be used as a useful indicator to histologically distinguish ameloblastic carcinoma from aggressive multicystic solid AME; whereas, in our study, we observed high OCT4 expression in 23 cases of solid AME. However, we cannot fail to consider the differences found when comparing the methodologies used.

OCT4 was also observed in epithelial and mesenchymal components during tooth development and is believed to participate in the ameloblast differentiation process (*Kero et al., 2014*). In both the AME samples and the AME-hTERT strain, OCT4 staining was predominantly nuclear. According to *Monroy et al. (2018)*, the presence of OCT4 in the nucleus is linked to cellular "stemness", which indicates that in ameloblastoma the nuclear expression of this protein may be related to tumour cell survival.

NANOG was found expressed in the tumour parenchyma of the tissue samples of AME with cytoplasmic and nuclear marking, as well as in the AME-hTERT lineage, in this study. The epithelial expression of NANOG has been described in head and neck cancers (*Lee et al., 2015*; *Rodrigues et al., 2017*). Its expression was recently investigated in the stroma of odontogenic lesions, including AME, by *Chacham et al. (2020)*, in which the presence of mesenchymal cells positive for NANOG was verified. Thus far, the present study is the only one to observe the expression of this protein in AME parenchyma.

In general, the expression pattern identified for SOX-2, NANOG and OCT4 is in accordance with their biological functions as transcription factors. The literature indicates that these markers are crucial transcription factors that are capable of allowing cancer cells to obtain properties similar to those of SCs (*Yu & Cirillo, 2020*). CSCs, in turn, manifest properties similar to those of SCs. These properties include the oncogenic reprogramming of different self-renewal genes, presenting characteristics of immortality, which persist in tumours, usually in nests, representing the source of expansion of growth and tumour maintenance, metastasis formation and tumour recurrence (*Liu et al., 2021*).

There are difficulties in the literature regarding naming the AME neoplastic cells that express these proteins. It seems inappropriate to call them SCs, given the various genetic and proteomic alterations described in AME neoplastic cells (*Gomes et al., 2010*). Even

more misleading would be to call them CSCs. The term "tumour stem cells" seems to be the most appropriate, since describing them as cells that have characteristics of SCs (*Monroy et al., 2018*) or cells that express SC (*Chacham et al., 2020*) markers seems somewhat vague.

*Rodda et al. (2005)*, showed, using chromatin immunoprecipitation, that OCT4 and SOX-2 interact synergistically and bind to NANOG in human and live mouse embryonic stem cells, driving the expression of target genes related to pluripotency. According to *Boyer et al. (2005)*, transcription factors together bind to the promoters of their own genes, forming interconnected autoregulatory loops and an autoregulatory network that is capable of providing advantages for the SCs, which are important for cell survival, stability and tumour progression. In comparison to the non-tumour epithelium, we observed that the expression levels of SOX-2, NANOG and OCT4 were increased in AME parenchyma, suggesting that these molecules may be involved in the pathogenesis of AME. It is noteworthy that in the literature, there are no studies that jointly assess the expression of these three markers both in the parenchyma of this neoplasm, as well as in cell lines originating from AME.

It is worth highlighting the need for further studies, such as mechanistic assays, which can suppress the expression of stem cell biomarkers and see the influence of this lock, given the limitations that the immunohistochemical study and the sample size may have. Furthermore, some steps in the use of the method used (IHC) aim to increase the specificity of the primary antibody, such as blocking with BSA and using a positive control. However, this does not guarantee the full specificity of the antibody used.

## CONCLUSIONS

Based on the obtained results, the high expression of SOX-2, NANOG and OCT4 markers in AME neoplastic cells using immunohistochemistry, and in the AME-hTERT cell line using immunofluorescence, was verified. The methods used confirm the presence and probable participation of these proteins in the origin and progression of AME. It is suggested that this tumour has cells with characteristics of CSs that could be related to the progression and recurrence of this odontogenic tumour.

### Funding

The authors received funding from the National Council for Scientific and Technological Development, CNPq (Grants #305493/2018-3 and #435644/2018-1 awarded to Ricardo Alves Mesquita; Grant 307584/2018-6 awarded to Rommel T J Ramos; and Grant #429423/2018-7 awarded to João de Jesus Viana Pinheiro). The funders had no role in study design, data collection and analysis, decision to publish, or preparation of the manuscript.

## Grant Disclosures

The following grant information was disclosed by the authors:

National Council for Scientific and Technological Development, CNPq: #305493/2018-3; #435644/2018-1; #307584/2018-6; #429423/2018-7.

## Competing Interests

Rommel Thiago Jucá Ramos is an Academic Editor for PeerJ.

## Author Contributions

- Karolyny Martins Balbinot conceived and designed the experiments, performed the experiments, analyzed the data, prepared figures and/or tables, authored or reviewed drafts of the article, and approved the final draft.
- Felippe José Almeida Loureiro conceived and designed the experiments, authored or reviewed drafts of the article, and approved the final draft.
- Giordanna Pereira Chemelo performed the experiments, authored or reviewed drafts of the article, and approved the final draft.
- Ricardo Alves Mesquita conceived and designed the experiments, authored or reviewed drafts of the article, and approved the final draft.
- Aline Maria Pereira Cruz Ramos analyzed the data, authored or reviewed drafts of the article, and approved the final draft.
- Rommel Thiago Jucá Ramos analyzed the data, prepared figures and/or tables, and approved the final draft.
- Artur Luiz da Costa da Silva analyzed the data, authored or reviewed drafts of the article, and approved the final draft.
- Sílvio Augusto Fernandes de Menezes conceived and designed the experiments, authored or reviewed drafts of the article, and approved the final draft.
- Maria Sueli da Silva Kataoka analyzed the data, authored or reviewed drafts of the article, and approved the final draft.
- Sergio de Melo Alves Junior analyzed the data, authored or reviewed drafts of the article, and approved the final draft.
- João de Jesus Viana Pinheiro conceived and designed the experiments, analyzed the data, prepared figures and/or tables, authored or reviewed drafts of the article, and approved the final draft.

## Human Ethics

The following information was supplied relating to ethical approvals (*i.e.*, approving body and any reference numbers):

Human Research Ethics Committee of the Health Sciences Institute of the Federal University of Pará—CEP/ICS/UFPA (CAAE: 30647720.6.0000.0018).

## Data Availability

The data is available at Figshare:

Balbinot, Karolyny (2022): OCT4.rarraw data. figshare. Figure. https://doi.org/10.6084/m9.figshare.19517200.v1.

Balbinot, Karolyny (2022): SOX-2.rarraw data. figshare. Figure. https://doi.org/10.6084/m9.figshare.19517263.v2.

Balbinot, Karolyny (2022): NANOG. figshare. Figure. https://doi.org/10.6084/m9.figshare.21507090.v1.

Balbinot, Karolyny (2022): Immunohistochemical quantification worksheets. figshare. Dataset. https://doi.org/10.6084/m9.figshare.21507096.v2.

Balbinot, Karolyny (2022): Immunofluorescence quantification worksheets. figshare. Dataset. https://doi.org/10.6084/m9.figshare.21507102.v1.

Balbinot, Karolyny (2022): Immunofluorescence of SOX-2, NANOG and OCT4 proteins. figshare. Figure. https://doi.org/10.6084/m9.figshare.21507117.v1.

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
