# Peer review of "Immunoexpression of stem cell markers SOX-2, NANOG AND OCT4 in ameloblastoma"

_PeerJ, doi:10.7717/peerj.14349_

## Round 0.1 · original submission · Major Revisions

When assessing your paper, the reviewers recommended that additional details are needed to fully support the results and conclusions, and for the manuscript to be suitable for publication in this journal.

Reviewer 1 ·

Basic reporting

It is recommended that authors review the manuscript for English language and grammar to make it suitable for international readership.
Discussion is too long and can be shortened to highlight the research findings in context of existing literature.
Raw data has not been shared, specifically for the data in Table 2 and Figure 3. More specifically, Table 2 should be presented in a graphical format instead of tabular to show the correlation between the proteins in individual samples.
Figure legends should be more detailed.

Experimental design

Research question is not very well defined, and the results do not support the conclusions that the authors make. This is specifically because the results have not been properly described. The authors need to conduct basic phenotypic/molecular biology assays to support their claim that there is a correlation between SOX-2, NANOG, OCT4.
Quantification of immunostaining of AME-hTert cells may be helpful to claim that expression of the proteins is where the authors think it is.

Validity of the findings

It appears that authors have conducted the experiments only once without any replicates. The results need to be more robust to claim that the three genes have a significant role in origin and progression of AME.

Reviewer 2 ·

Basic reporting

The manuscript titled "Immunoexpression of stem cell markers SOX-2, NANOG and OCT4 in ameloblastoma" by Balbinot et al describes expression of pluripotent embryonic stem cell markers in ameloblastoma. The language used in the manuscript is clear and unambiguous. I appreciate the authors for writing an elaborate introduction. But, I felt the introduction should be restructured to suit the data presented in the manuscript. The main problem with the introduction is over emphasis of stem cells and cancer stem cells. SOX-2, NANOG and OCT4 are embryonic stem cell markers but in tumors their role could be very different. Expression of these genes may not necessarily imply that those cells are either CSC or SCs. The introduction therefore appears misleading. For example, (lines 85-88) give an impression that the authors were assuming that expression of these three genes in AME tumors indicate the pluripotency of cancerous cells - which is inaccurate.

I suggest authors to emphasize that SOX-2, NANOG and OCT4 have oncogenic roles in several cancers and explain 1) what cancer phenotypes they influence and 2) their association with patients characteristics and prognosis and therapeutic response. Such information will help the reader understand the significance of studying their expression using IHC in tumors.

I appreciate authors for the quality of IHC images. The staining looks very clean.

Experimental design

Authors used no primary antibody as negative control in IHC. This does not rule the possibility of nonspecific binding of primary antibody. Please add a statement on this in discussion. Limitations statement in discussion is necessary and discuss sample size limitations.

Validity of the findings

1) Line 170: The male gender is more prevalent...... Is it statistically significant?
2) Line 189: A r values of 0.45 indicates a very weak correlation. Please note that r (not p value) is the appropriate metric for assessing if there is a correlation between the variables. In this case the r value shows that the correlation is very minimal. Of note, the corresponding p value in this analysis indicate the measured r value is not an error. It does NOT imply that there is a positive correlation.

3) Similarly, for linear regression, an r2 value of 0.2 indicates that there is no linear relationship. Here the p value indicates accuracy of the value of coefficient of the predictors. Please note that the p value doe NOT indicate that there is a linear relationship.

4) it is not clear what did authors find from transcriptome data. Lines 198-203 gave an impression that the authors used the three gene names as input for string database. I did not understand how gene expression profiling performed using Ion Torrent machine was analyzed and what results were obtained.
5) I suggest authors to remove data shown in figure 3. This is not new finding. It has been known that the three proteins interact with each other.

---

## Round 0.2 · Minor Revisions

Thank you for revising the manuscript. Reviewer #2 has raised technical concerns that should be addressed for manuscript clarity.

Reviewer 1 ·

Basic reporting

Authors have made great efforts to satisfactorily address the comments.

Experimental design

Authors have made great efforts to satisfactorily address the comments.

Validity of the findings

Authors have made great efforts to satisfactorily address the comments.

Reviewer 2 ·

Basic reporting

No comment

Experimental design

The main strength of this manuscript is showing higher expressions of SOX-2, NANOG and OCT4 in AME. Though sample size is small, the cohort includes multiple histological types of the disease. This study sets premise for conducting future studies comprising larger cohorts to evaluate the clinical significance of these genes in AME.
You do not have appropriate control for your IHC to rule out that your antibodies are not binding to nonspecific target. Using BSA may help to reduce nonspecific recognition of antigen, but you can’t prove it unless you have tissue wherein the given protein does not express as a negative control. Please acknowledge this in discussion.

Validity of the findings

Please note that if two different genes express in same condition or if their expression is correlated, it never reinforces that they physically interact. Authors wrongly assume that co-expression indicates evidence of physical interaction. Often, proteins interacting will not show linear correlation of expression and vice versa. How does existence of physical interaction two proteins suggest that weak correlation is due to small size. Indeed, weak correlation could be due to small sample size, but it is independent of interaction of two proteins. They show that all three proteins are interacting with each other. If authors believe that interaction reinforces correlation of expression, why all three proteins expression patterns are not correlating with each other’s. Therefore lines 341 -345 in discussion do not make sense. This also brings up another question “what is the need to show interaction between the three proteins?”. What do you want the reader to understand from it?

The bioinformatic analysis and the corresponding results are the weakest part of this manuscript, and it shows that author’s lack of understanding of such concepts. The methodological description in original submission was superficial. They did not give a rationale for using RNAseq. They did not state what was the sample used for RNAseq. Authors stated in response that in the original submission they simplified methods and in the revised manuscript they expanded it. All they did was mentioning they used GFOLD for read counting. GFOLD algorithm is used to measure fold change when there are no replicates. But it essentially requires two samples to compare. It can’t be used for ‘read counting’ as claimed by the authors. What about QC, read filtering and mapping? In addition, PRJNA860081 has just one sample which is human ameloblastoma non-immortalized (AME-1). How do you calculate fold change with just one sample? RNAseq data were shown only in table 2 and its corresponding text. Table 2 shows number reads mapped to each gene. What did you use GFOL for? You showed that out of all 6 genes shown in table 2, SALL4 has maximum number of reads (24 reads) mapped to it, which is miniscule number given that you have 37.7 million total reads per sample. Out of 6 genes, 4 of them have 2 or less reads mapped. This suggests that all 6 genes have low expression level. Therefore, your claims are not supported by the data presented, and overdrawn and suffer from misinterpretations. Therefore, I feel that this paper will read well without any bioinformatic analysis including protein interactions and transcriptome analysis.

---

## Round 0.3 · accepted · Accept

Thank you for submitting the manuscript.